# NEURAL CLUSTERING PROCESSES

## ABSTRACT

Mixture models, a basic building block in countless statistical models, involve latent random variables over discrete spaces, and existing posterior inference methods can be inaccurate and/or very slow. In this work we introduce a novel deep learning architecture for efficient amortized Bayesian inference over mixture models. While previous approaches to amortized clustering assumed a fixed or maximum number of mixture components and only amortized over the continuous parameters of each mixture component, our method amortizes over the local discrete labels of all the data points, and performs inference over an unbounded number of mixture components. The latter property makes our method natural for the challenging case of nonparametric Bayesian models, where the number of mixture components grows with the dataset. Our approach exploits the exchangeability of the generative models and is based on mapping distributed, permutation-invariant representations of discrete arrangements into varying-size multinomial conditional probabilities. The resulting algorithm parallelizes easily, yields iid samples from the approximate posteriors along with a normalized probability estimate of each sample (a quantity generally unavailable using Markov Chain Monte Carlo) and can easily be applied to both conjugate and non-conjugate models, as training only requires samples from the generative model. We also present an extension of the method to models of random communities (such as infinite relational or stochastic block models). As a scientific application, we present a novel approach to neural spike sorting for high-density multielectrode arrays.

## 1 INTRODUCTION

Mixture models (or equivalently, probabilistic clustering models) are a staple of statistical modelling in which a discrete latent variable is introduced for each observation, indicating its mixture component identity. Popular inference methods in these models fall into two main classes. When exploring the full posterior is crucial (e.g. there is irreducible uncertainty about the latent structure or many separate local optima exist), the method of choice is Markov Chain Monte Carlo (MCMC) (Neal, 2000; Jain & Neal, 2004). This method is asymptotically accurate but time-consuming, with convergence that is difficult to assess. Models whose likelihood and prior are non-conjugate are particularly challenging, since in general in these cases the model parameters cannot be marginalized and must be kept as part of the state of the Markov chain. Alternatively, variational methods (Blei & Jordan, 2004; Kurihara et al., 2007; Hughes et al., 2015) are typically much faster but do not come with accuracy guarantees.

As an alternative to MCMC and variational approaches, in recent years there has been steady progress on amortized inference methods, and such is the spirit of this work. Concretely, we propose a novel technique to perform amortized approximate posterior inference over discrete latent variables in mixture models. The basic idea is to use neural networks to express posteriors in the form of multinomial distributions (with varying support) in terms of fixed-dimensional, distributed representations that respect the permutation symmetries imposed by the discrete variables. A major advantage of our architecture, compared to previous approaches to amortized clustering, is its ability to handle an arbitrary number of clusters. This makes the method a natural choice for nonparametric Bayesian models, such as Dirichlet process mixture models (DPMM), and their extensions, where the number of components, a measure of the model complexity, is inferred as a posterior random variable; see (Rodriguez & Mueller, 2013) for a recent overview. Moreover, the method can be applied to both conjugate and non-conjugate models.

The term 'amortization' refers to the process of investing computational resources to train a model that is later used for very fast posterior inference (Gershman & Goodman, 2014). Concretely, the amortized approach learns a parametrized function $q_\theta(z|x)$ that approximates $p(z|x)$ for any $x$; learning the model parameters $\theta$ may be computationally challenging, but once $\theta$ is in hand then evaluating $q_\theta(z|x)$ for new data $x$ is fast.

The amortized inference literature can be coarsely divided into two approaches. On one side, the variational autoencoder approach (Kingma & Welling, 2013), with roots in the wake-sleep algorithm (Hinton et al., 1995), learns $q_\theta(z|x)$ along with the generative model $p_\phi(x|z)$. While $p(z)$ is usually a known simple distribution, for discrete latent variables backpropagation cannnot be performed through them, and special approaches have been developed for those cases (Mnih & Rezende, 2016; Jang et al., 2016; Maddison et al., 2016).

Our work corresponds to the alternative case: a generative model $p(x, z)$ is postulated , and posterior inference is the main focus of the learning phase. Amortized methods in this case usually involve a degree of specialization to the particular generative model of interest. Examples include methods developed for Bayesian networks (Stuhlmüller et al., 2013), sequential Monte Carlo (Paige & Wood, 2016), probabilistic programming (Ritchie et al., 2016; Le et al., 2016), neural decoding (Parthasarathy et al., 2017) and particle tracking (Sun & Paninski, 2018). Our work is specialized to the case where the latent variables are discrete and their range is not fixed beforehand.

In the approach we present, after training the neural architecture using labeled samples from a particular generative model, we can obtain independent, parallelizable, approximate posterior samples of the discrete variables for any new set of observations of arbitrary size, with no need for expensive MCMC steps. These samples can be used (i) to compute approximate expectations, (ii) as high quality importance samples, or (iii) as independent Metropolis-Hastings proposals.

In Section 2 we study amortized mixtures and in Section 3 we review related works. In Section 4 we discuss quantitative evaluations of the new method. In Section 5 we present an extension of the method to random community graph models. We close in Section 6 with a neuroscientific application of this method to spike sorting for high-density multielectrode probes.

## 2 AMORTIZING MIXTURE MODELS

We start by presenting mixture models from the perspective of probabilistic models for clustering (McLachlan & Basford, 1988). The latter introduce random variables $c_i$ denoting the cluster number to which the data point $x_i$ is assigned, and assume a generating process of the form

$$\alpha_1, \alpha_2 \sim p(\alpha)$$
$$N \sim p(N)$$
$$c_1 \ldots c_N \sim p(c_1, \ldots, c_N | \alpha_1)$$
$$\mu_1 \ldots \mu_K | c_{1:N} \sim p(\mu_1, \ldots \mu_K | \alpha_2)$$
$$x_i \sim p(x_i | \mu_{c_i}) \quad i = 1 \ldots N$$

Here $\alpha_1, \alpha_2$ are hyperparameters. The number of clusters $K$ is a random variable, indicating the number of distinct values among the sampled $c_i$'s, and $\mu_k$ denotes a parameter vector controlling the distribution of the $k$-th cluster (e.g., $\mu_k$ could include both the mean and covariance of a Gaussian mixture component). We assume that the priors $p(c_{1:N}|\alpha_1)$ and $p(\mu_{1:K}|\alpha_2)$ are exchangeable,

$$p(c_1, \ldots, c_N | \alpha_1) = p(c_{\sigma_1}, \ldots, c_{\sigma_N} | \alpha_1),$$

where $\{\sigma_i\}$ is an arbitrary permutation of the indices, and similarly for $p(\mu_{1:K}|\alpha_2)$. Our interest in this work is in cases where $K$ can take any value $K \leq N$, such as the Chinese Restaurant Process (CRP), or its Pitman-Yor generalization. Of course, our methods will also work for models with $K < B$ with fixed $B$, such as Mixtures of Finite Mixtures (Miller & Harrison, 2018).

Given $N$ data points $\mathbf{x} = \{x_i\}$, we would like to draw independent samples from the posterior

$$p(c_{1:N}|\mathbf{x}) = p(c_1|\mathbf{x})p(c_2|c_1, \mathbf{x}) \ldots p(c_N|c_{1:N-1}, \mathbf{x}). \tag{1}$$

Note that $p(c_1 = 1|\mathbf{x}) = 1$, since the first data point is always assigned to the first cluster. While we might also be interested in the hidden variables $\alpha_1, \alpha_2, \mu_k$, the reason to focus on the discrete

variables $c_i$'s is that given samples from them, it is generally relatively easy to obtain posterior samples from $p(\alpha_1|c_{1:N})$ and $p(\mu_k, \alpha_2|\mathbf{x}, c_{1:N})$.

We would like to model all the factors in (1) in a unified way, with a generic factor given by

$$p(c_n|c_{1:n-1}, \mathbf{x}) = \frac{p(c_1 \dots c_n, \mathbf{x})}{\sum_{c'_n=1}^{K+1} p(c_1 \dots c'_n, \mathbf{x})}. \tag{2}$$

Here we assumed that there are $K$ unique values in $c_{1:n-1}$, and therefore $c_n$ can take $K+1$ values, corresponding to $x_n$ joining any of the $K$ existing clusters, or forming its own new cluster.

We are interested in approximating (2):

$$p(c_n|c_{1:n-1}, \mathbf{x}) \approx q_\theta(c_n|c_{1:n-1}, \mathbf{x}), \tag{3}$$

where $q_\theta$ is parameterized by a flexible model such as a neural network that takes as inputs $(c_{1:n-1}, \mathbf{x})$, then extracts features and combines them nonlinearly to output a probability distribution on $c_n$. Critically, we will design the network to enforce the highly symmetric structure of the lhs of (3).

To make this symmetric structure more transparent, and in light of the expression (2), let us consider the joint distribution of the assignments of the first $n$ data points,

$$p(c_1, \dots, c_n, \mathbf{x}). \tag{4}$$

A neural representation of this quantity should respect the permutation symmetries imposed on the $x_i$'s by the values of $c_{1:n}$. Therefore, our first task is to build permutation-invariant representations of the observations $\mathbf{x}$. The general problem of constructing such invariant encodings was discussed recently in (Zaheer et al., 2017); to adapt this approach to our context, we consider three distinct permutation symmetries:

- **Permutations within a cluster:** (4) is invariant under permutations of $x_i$'s in the same cluster. For each of the $K$ clusters that have been sampled so far, we define the encoding

$$H_k = \sum_{i:c_i=k} h(x_i) \quad h : \mathbb{R}^{d_x} \to \mathbb{R}^{d_h} \tag{5}$$

which is clearly invariant under permutations of $x_i$'s in the same cluster. In general $h$ is an encoding function we learn from data, unless $p(x|\mu)$ belongs to an exponential family and the prior $p(c_{1:N})$ is constant, as shown in Appendix A.

- **Permutations between clusters:** (4) is invariant under permutations of the cluster labels. In terms of the within-cluster invariants $H_k$, this symmetry can be captured by

$$G = \sum_{k=1}^{K} g(H_k), \quad g : \mathbb{R}^{d_h} \to \mathbb{R}^{d_g}. \tag{6}$$

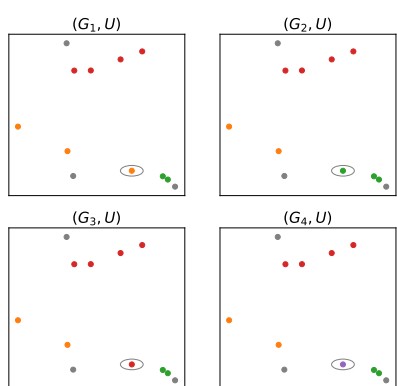

Figure 1: **Encoding cluster labels.** The colored points have fixed labels $c_{1:n-1}$, forming $K = 3$ clusters. The four possible labels for the circled point give four encoding vectors $G_k$, while the vector $U$ encodes the 3 gray unlabeled points (Best seen in color).

- **Permutations of the unassigned data points:** (4) is also invariant under permutations of the $N - n$ unassigned data points. This can be captured by

$$U = \sum_{i=n+1}^{N} u(x_i), \quad u : \mathbb{R}^{d_x} \to \mathbb{R}^{d_u}. \tag{7}$$

Note that $G$ and $U$ provide fixed-dimensional, symmetry-invariant representations of the assigned and non-assigned data points, respectively, for any values of $N$ and $K$. Encodings of this form were shown in (Zaheer et al., 2017) to lead to arbitrarily accurate approximations of symmetric functions.

Figure 2: **Architecture of the Neural Clustering Process.** The full model is composed by the deep networks $h, g, u, f$. *Left:* After assigning the cluster labels $c_{1:n-1}$, each possible discrete value $k$ for $c_n$ gives a different symmetry-invariant encoding of $x_{1:n}$ into the vector $G_k$, using the functions $h$ and $g$. The remaining, yet-unassigned points $x_{n+1:N}$ are encoded by $u$ and summed into the vector $U$. *Right:* Each pair $G_k, U$ is mapped by $f$ into a real number (logit), which in turn is mapped into the multinomial distribution $q_\theta(c_n|c_{1:n-1}, \mathbf{x})$ via a variable-input softmax.

## 2.1 THE VARIABLE-INPUT SOFTMAX

After assigning values to $c_{1:n-1}$, each of the $K+1$ possible values for $c_n$ corresponds to $h(x_n)$ appearing in one particular $H_k$ in (5), and yields a separate vector $G_k$ in (6). See Figure 1 for an example. In terms of the $G_k$'s and $U$, we propose to model (2) as

$$q_\theta(c_n = k|c_{1:n-1}, \mathbf{x}) = \frac{e^{f(G_k, U)}}{\sum_{k'=1}^{K+1} e^{f(G_{k'}, U)}} \qquad k = 1 \ldots K+1, \tag{8}$$

where we have introduced a new real-valued function $f$. In other words, each value of $c_n$ corresponds to a different channel through which the encoding $h(x_n)$ flows to the logit value $f$, as shown in Figure 2. Note that $k = K+1$ corresponds to $c_n$ forming its own new cluster with $H_k = h(x_n)$.

The softmax (8) differs from its usual form in, e.g., classification networks, where a fixed number of categories receive their logit values $f$ from the fixed-size final layer of an MLP. In our case, the discrete identity of each logit is determined by the neural path that the input $h(x_n)$ takes to $G$, thus allowing a flexible number of categories.

In eq. (8), $\theta$ denotes the parameters in the functions $h, g, u$ and $f$, which we represent with neural networks. By storing and updating $G$ and $U$ for successive values of $n$, the computational cost of a full i.i.d. sample of $c_{1:N}$ is $O(NK)$, the same as a single Gibbs sweep. See Algorithm 1 for details; we term this approach the Neural Clustering Process (NCP). It is relatively easy to run hundreds of copies of Algorithm 1 in parallel on a GPU, with each copy yielding a different set of samples $c_{1:N}$.[1]

## 2.2 THE OBJECTIVE FUNCTION

In order to learn the parameters $\theta$ of the neural networks, we use stochastic gradient descent to minimize the expected KL divergence,

$$\mathbb{E}_{p(N)p(\mathbf{x})} D_{\mathrm{KL}}(p(c|\mathbf{x}) \| q_\theta(c|\mathbf{x})) = -\mathbb{E}_{p(N)} \mathbb{E}_{p(c_{1:N}, \mathbf{x})} \left[ \sum_{n=2}^{N} \log q_\theta(c_n|c_{1:n-1}, \mathbf{x}) \right] + \text{const.} \tag{9}$$

Samples from $p(c_{1:N}, \mathbf{x})$ are obtained from the generative model, irrespective of the model being conjugate. If we can take an unlimited number of samples from the generative model, we can potentially train a neural network to approximate $p(c_n|c_{1:n-1}, \mathbf{x})$ arbitrarily accurately. Note that the gradient here acts only on the variable-input softmax term $q_\theta$, not $p(c, \mathbf{x})$, so there is no problem of backpropagating through discrete variables (Jang et al., 2016; Maddison et al., 2016).

---

[1]A Pytorch implementation of the algorithm is available at `https://bit.ly/2lkGJ1b`

**Algorithm 1** $O(NK)$ Neural Clustering Process Sampling

---

1: $h_i \leftarrow h(x_i), u_i \leftarrow u(x_i) \qquad i = 1 \ldots N$ {Notation}
2: $U \leftarrow \sum_{i=2}^{N} u_i$ {Initialize unassigned set}
3: $H_1 \leftarrow h_1, G \leftarrow g(H_1), K \leftarrow 1, c_1 \leftarrow 1$ {Create first cluster with $x_1$}
4: **for** $n \leftarrow 2 \ldots N$ **do**
5: $\quad U \leftarrow U - u_n$ {Remove $x_n$ from unassigned set}
6: $\quad H_{K+1} \leftarrow 0$ {We define $g(0) = 0$}
7: $\quad$ **for** $k \leftarrow 1 \ldots K + 1$ **do**
8: $\qquad G_k \leftarrow G + g(H_k + h_n) - g(H_k)$ {Add $x_n$ to cluster $k$}
9: $\qquad q_k \leftarrow e^{f(G_k, U)}$
10: $\quad$ **end for**
11: $\quad q_k \leftarrow q_k / \sum_{k'=1}^{K+1} q_{k'}, \quad c_n \sim q_k$ {Normalize probabilities and sample assignment}
12: $\quad$ **if** $c_n = K + 1$ **then**
13: $\qquad K \leftarrow K + 1$
14: $\quad$ **end if**
15: $\quad G \leftarrow G - g(H_{c_n}) + g(H_{c_n} + h_n)$ {Add point $x_n$ to sampled cluster $c_n$}
16: $\quad H_{c_n} \leftarrow H_{c_n} + h_n$
17: **end for**
18: Return $c_1 \ldots c_N$

---

### 2.3 TWO EXAMPLES

**Clustering in 2D Gaussian models:** The generative model is

$$
\begin{aligned}
\alpha &\sim \mathrm{Exp}(1) \quad c_{1:N} \sim \mathrm{CRP}(\alpha) & \mu_k &\sim N(0, \sigma_\mu^2 \mathbf{1}_2) \quad k = 1 \ldots K \\
N &\sim \mathrm{Uniform}[5, 100] & x_i &\sim N(\mu_{c_i}, \sigma^2 \mathbf{1}_2) \quad i = 1 \ldots N
\end{aligned}
\tag{10}
$$

where CRP stands for the Chinese Restaurant Process, with concentration parameter $\alpha$, $\sigma_\mu = 10$, and $\sigma = 1$. Figure 3 shows that the NCP captures the posterior uncertainty inherent in clustering this data. Note that since the generative model is an analytical known distribution, there is no distinction here between training and test sets.

**Clustering of MNIST digits:** We consider next a DPMM over MNIST digits, with generative model

$$
\begin{aligned}
\alpha &\sim \mathrm{Exp}(1) \quad c_{1:N} \sim \mathrm{CRP}_{10}(\alpha) & l_k &\sim \quad \mathrm{Unif}[0, 9] - \text{without replacement.} \quad k = 1 \ldots K \\
N &\sim \mathrm{Uniform}[5, 100] & x_i &\sim \quad \mathrm{Unif}[\text{MNIST digits with label } l_{c_i}] \quad i = 1 \ldots N
\end{aligned}
$$

where $\mathrm{CRP}_{10}$ is a Chinese Restaurant Process truncated to up to 10 clusters, and $d_x = 28 \times 28$. Training was performed by sampling $x_i$ from the MNIST training set. Figure 4 shows posterior samples for a set of digits from the MNIST test set, illustrating how the estimated model correctly captures the shape ambiguity of some of the digits. Note that in this case the generative model has no analytical expression (and therefore is non-conjugate), but this presents no problem; a set of labelled samples is all we need for training. See Appendix F for details of all the network architectures used.

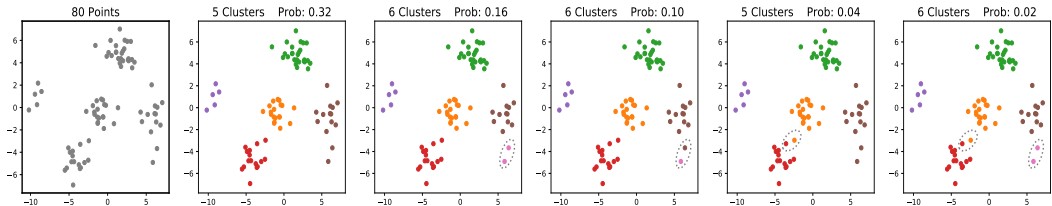

Figure 3: **Mixture of 2D Gaussians:** Given the observations in the leftmost panel, we show samples from the NCP posterior. Note that less-reasonable samples are assigned lower probability by the NCP. The dotted ellipses indicate assignments which differ from the first, highest-probability sample. Our GPU implementation gives thousands of such samples in a fraction of a second. (Best seen in color.)

Figure 4: **NCP trained on MNIST clusters.** The top row shows 20 images from the MNIST test set. The five rows below show five samples of $c_{1:20}$ from the NCP posterior. Note that each sample captures some ambiguity suggested by the form of particular digits.

## 3 RELATED WORKS

Permutation-invariant neural architectures have been explored recently in (Ravanbakhsh et al., 2017; Korshunova et al., 2018; Lee et al., 2018; Bloem-Reddy & Teh, 2019; Wagstaff et al., 2019). The representation of a set via a sum (or mean) of encoding vectors was also used in (Guttenberg et al., 2016; Ravanbakhsh et al., 2016; Edwards & Storkey, 2017; Zaheer et al., 2017; Garnelo et al., 2018a;b).

Most works on neural network-based clustering focus on learning features as inputs to traditional clustering algorithms, as reviewed in (Du, 2010; Aljalbout et al., 2018; Min et al., 2018). The works closest to ours are (Le et al., 2016) and (Lee et al., 2018). Both present techniques for amortized inference of mixtures of Gaussians, so it is instructive to compare them in detail to our approach.

The work (Le et al., 2016) studies amortized inference of a variable number of latent variables generated during the trace of a general sequential probabilistic program. For the case of a mixture of 2D Gaussians with a latent random number of components, a 2D histogram image of binned observations is fed to a convolutional network whose output enters into a recurrent neural network with a fixed-sized softmax output layer to estimate the number of clusters. The network also outputs the means and covariances of each cluster.

The work (Lee et al., 2018) presents Set Transformer, an attention-based architecture that improves the simple sum-based set encoding that we used above. In their 2D Gaussian clustering application, the number of components is fixed beforehand, and inference is made only on the cluster parameters.

These approaches have several limitations compared to ours. First, the number of clusters is upper bounded by the size of the softmax layer (Le et al., 2016) or fixed (Lee et al., 2018). Second, the models perform inference on the continuous parameters $\mu_k$, but not on the discrete labels of each data point. Finally, in (Le et al., 2016), the use of a convnet on a 2D histogram to determine the number of clusters does not scale to higher dimensional data due to the curse of dimensionality. In Table 1 we summarize the comparison between the three approaches.

| Property | NCP | Program Compilation | Set Transformer |
|---|---|---|---|
| Number of mix. components | Arbitrary | Bounded | Fixed |
| Amortizes discrete labels | Yes | No | No |
| Amortizes component parameters | No | Yes | Yes |
| Scales to high dimensional data | Yes | No | Yes |

Table 1: **Comparing amortized approaches to Gaussian mixtures.** We compare NCP with Program Compilation (Le et al., 2016) and Set Transformer (Lee et al., 2018), two previous approaches to amortized mixtures of Gaussians. Note however that NCP can be applied to any mixture model.

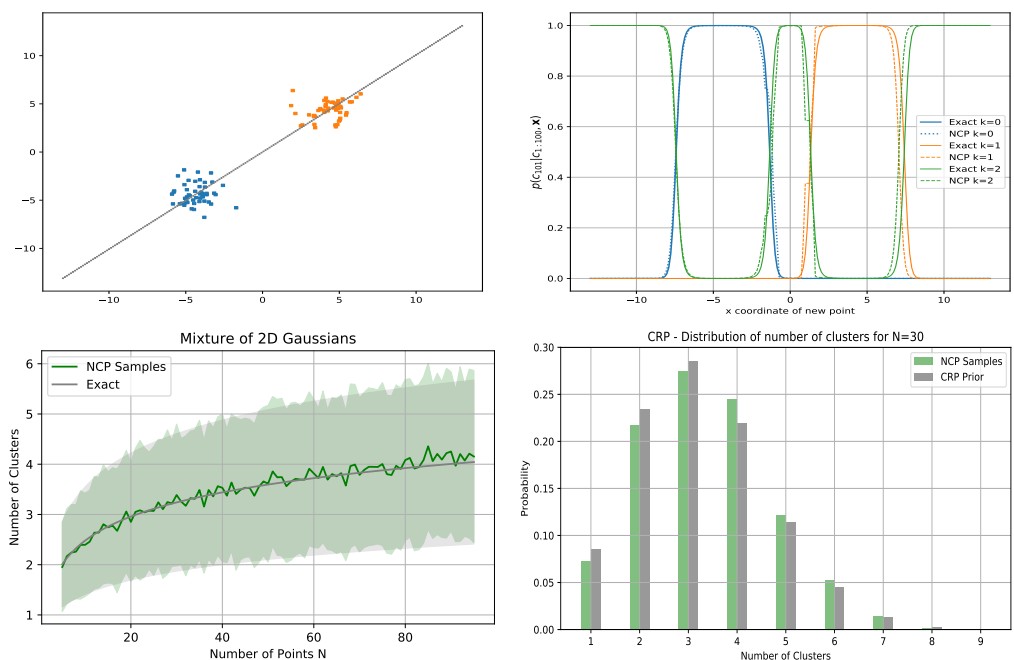

Figure 5: **Quantitative Evaluations.** *Upper left:* Two 2D clusters of 50 points each ($k = 0, 1$) and a line over possible locations of a 101st last point. *Upper right:* Assuming the 2D model from (10), the posterior $p(c_{101}|c_{1:100}, \mathbf{x})$ can be computed exactly, and we compare it to the NCP estimate as a function of the horizontal coordinate of $x_{101}$, as this point moves over the gray line on the upper left panel. *Geweke's Tests. Lower left:* The curves compare the exact mean ($\pm$ one std.) of the number of clusters $K$ for different $N$'s from the CRP prior (with $\alpha = 0.7$), with sampled estimates using equation (11). *Lower right:* Similar comparison for the full histogram of $K$ for $N = 30$ points.

## 4 EXPECTATIONS, EVALUATIONS AND DIAGNOSTICS

Samples from the NCP can be used to compute approximate expectations. If the interest is in asymptotically exactness, the samples can be used as self-normalized importance samples, $\mathbb{E}[f(c)] = \sum_{i=1}^{M} f(c^{(i)}) w_i / \sum_{i=1}^{M} w_i$ where $w_i = p(\mathbf{x}, c^{(i)})/q_\theta(c^{(i)}|\mathbf{x})$. Alternatively, the samples can be used as proposals in Metropolized independent sampling (Liu, 1996). Of course, in both cases the variance of the estimated expectations will be lower when the NCP posterior is closer to the true posterior.

The examples presented in Sec. 2.3 provide strong qualitative evidence that our approximations to the true posterior distributions in these models are capturing the uncertainty inherent in the observed data. But we would like to go further and ask quantitatively how well our approximations match the exact posterior. Unfortunately, for sample sizes much larger than $N = O(10)$ it is impossible to compute the exact posterior in these models. Nonetheless, there are several quantitative metrics we can examine to check the accuracy of the model output.

**Global symmetry from exchangeability:** Our results relied on $p(c_{1:N}|\alpha_1)$ being exchangeable, which in turn implies exchangeability of the joint posterior (1). But this is not explicit in the rhs of (1), where a particular order is chosen for the expansion. If our model learns the conditional probabilities correctly, this symmetry should be (approximately) satisfied, and this can be monitored during training, as we show in Appendix C.

**Estimated vs. Analytical Probabilities:** Some conditional probabilities can be computed analytically and compared with the estimates output by the network; in the example shown in Figure 5, upper-right, the estimated probabilities are in close agreement with their exact values.

**Geweke's Tests:** A popular family of tests that check the correctness of MCMC implementations (Geweke, 2004) can also be applied in our case: verify the (approximate) identity between the

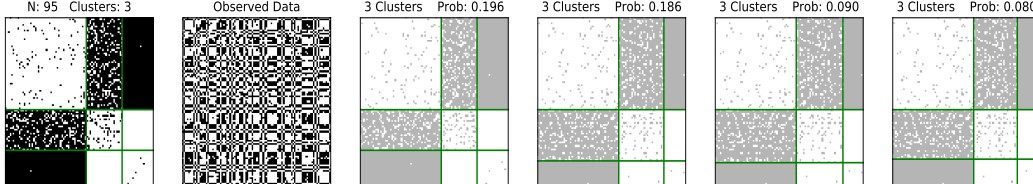

Figure 6: **Community Detection with Neural Block Processes.** The model is a single-type Infinite Relational Model (Kemp et al., 2006; Xu et al., 2006), with a CRP prior with $\alpha = 0.7$. The entries in each block are Bernoulli samples, with a block parameter sampled from a $\text{Beta}(0.2, 0.2)$ prior. From left to right: (i) the original block structure, sampled from the generative model, (ii) the observed random permutation of rows and columns, (iii) four samples from the NBP posterior, along with their estimated probabilities. Each sample from the posterior here corresponds to a plausible partition.

prior $p(c_{1:N})$ and

$$q_\theta(c_{1:N}) \equiv \int d\mathbf{x} \, q_\theta(c_{1:N}|\mathbf{x}) \, p(\mathbf{x}) \,, \tag{11}$$

where $p(\mathbf{x})$ is the marginal from the generative model. Figure 5 shows such a comparison for the 2D Gaussian DPMM from Section 2.3, showing excellent agreement.

**NCP vs. MCMC:** NCP has some advantages over MCMC approaches. First, it gives a probability estimate for each sample, in general unavailable in MCMC. Secondly, NCP enjoys higher efficiency, due to parallelization of iid samples. For example, in the Gaussian 2D example in eq.(10), in the time a collapsed Gibbs sampler produces one (correlated) sample, our GPU-based method produces more than 100 iid approximate samples. Finally, NCP does not need a burn-in period.

**NCP vs. Variational Inference:** In Section 6, we compare NCP with a variational approach on clustering neural spikes. For 2000 spikes, the variational approach returned one clustering estimate in 0.76 secs., but does not properly handle the uncertainty about the number of clusters. NCP produced 150 clustering configurations in 10 secs., efficiently capturing clustering uncertainty. In addition, the variational approach requires a preprocessing step that projects the samples to lower dimensions, whereas NCP directly consumes the high-dimensional data by learning an encoder function $h$.

## 5 COMMUNITIES

As an extension, we consider now a similar prior as above over cluster labels, but the observation model is more challenging:

$$\alpha, N \sim p(\alpha), p(N) \qquad\qquad \phi_{k_1,k_2} \sim p(\phi|\beta) \quad k_1 \le k_2$$
$$c_1 \ldots c_N \sim p(c_1, \ldots, c_N|\alpha) \qquad\qquad x_{i,j} \sim \text{Bernoulli}(\phi_{c_i,c_j}), \quad i \le j, \quad i,j = 1 \ldots N$$

where $k_1, k_2 = 1 \ldots K$. Here $p(c_{1:n}|\alpha)$ can be any exchangeable prior, and the binary observations $x_{i,j}$ represent edges in a graph of $N$ vertices. We focus on the symmetric graph case here, so $\phi_{k_1,k_2} = \phi_{k_2,k_1}$ and $x_{i,j} \equiv x_{j,i}$. We use a Beta model for $p(\phi|\beta)$, but other choices are possible.

These models include stochastic block models (Holland et al., 1983; Nowicki & Snijders, 2001) and the single-type Infinite Relational Model (Kemp et al., 2006; Xu et al., 2006; Schmidt & Morup, 2013). Neural architectures for communities in graphs have been studied in (Chen et al., 2019) as a classification problem for every node over a fixed predetermined number of clusters.

We could proceed similarly to the clustering case, considering $N$ particles, each given by a row of the adjacency matrix $\mathbf{x}_i = (x_{i,1} \ldots x_{i,N})$. But we should be careful when encoding these particles. When values of $c_{1:n}$ are assigned, a generic encoding $h(\mathbf{x}_i)$ would ignore the permutation symmetries present among the components of $\mathbf{x}_i$, i.e., the columns of $x_{i,j}$, as a result of the $c_{1:n}$ assignments (the same three permutation symmetries discussed above for clustering models). Moreover, a fixed encoding $h(\mathbf{x}_i)$ cannot accommodate the arbitrary length $N$ of $\mathbf{x}_i$. In Appendix B we present an invariant encoding that respects all these requirements. We call our approach Neural Block Process (NBP). See Figure 6 for an example.

## 6   APPLICATION: SPIKE SORTING WITH NCP

Large-scale neural population recordings using multi-electrode arrays (MEA) are crucial for understanding neural circuit dynamics. Each MEA electrode reads the signals from many neurons, and each neuron is recorded by multiple nearby electrodes. As a key analysis step, spike sorting converts the raw signal into a set of spike trains belonging to individual neurons (Pachitariu et al., 2016; Chung et al., 2017; Jun et al., 2017; Lee et al., 2017; Chaure et al., 2018; Carlson & Carin, 2019). At the core of many spike sorting pipelines is a clustering algorithm that groups the detected spikes into clusters, each representing a putative neuron (Figure 7). However, clustering spikes can be challenging: (1) Spike waveforms form highly non-Gaussian clusters in spatial and temporal dimensions, and it is unclear what are the optimal features for clustering. (2) It is unknown *a priori* how many clusters there are. (3) Existing methods do not perform well on spikes with low signal-to-noise ratios (SNR) due to increased clustering uncertainty, and fully-Bayesian approaches proposed to handle this uncertainty (Wood & Black, 2008; Carlson et al., 2013) do not scale to large datasets.

To address these challenges, we propose a novel approach to spike clustering using NCP. We consider the spike waveforms as generated from a Mixture of Finite Mixtures (MFM) distribution (Miller & Harrison, 2018), which can be effectively modeled by NCP. (1) Rather than selecting arbitrary features for clustering, the spike waveforms are encoded with a convolutional neural network (ConvNet), which is learned end-to-end jointly with the NCP network to ensure optimal feature encoding. (2) Using a variable-input softmax function, NCP is able to perform inference on cluster labels without assuming a fixed or maximum number of clusters. (3) NCP allows for efficient probablistic clustering by GPU-parallelized posterior sampling, which is particularly useful for handling the clustering uncertainty of ambiguous small spikes. (4) The computational cost of NCP training can be highly amortized, since neuroscientists often sort spikes form many statistically similar datasets.

We trained NCP for spike clustering using synthetic spikes from a simple yet effective generative model that mimics the distribution of real spikes, and evaluated the spike sorting performance on labeled synthetic data, unlabeled real data and hybrid test data by comparing NCP against two other methods: (1) **vGMFM**, variational inference on Gaussian MFM (Hughes & Sudderth, 2013). (2) **Kilosort**, a state-of-the-art spike sorting pipeline described in Pachitariu et al. (2016). In Appendix D, we describe the dataset, neural architecture, and the training/inference pipeline of NCP spike sorting.

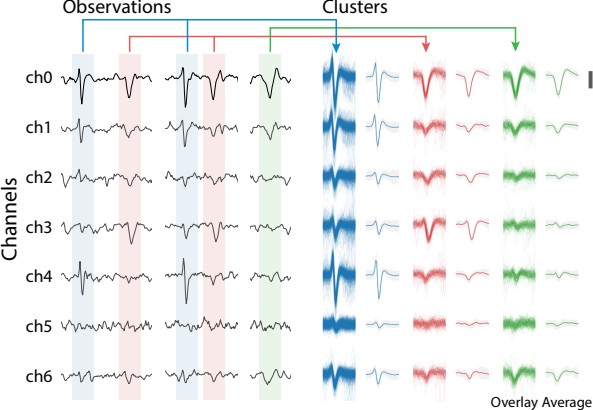

Figure 7: **Clustering multi-channel spike waveforms using NCP.** Each row is an electrode channel. Spikes with the same color belong to the same cluster. (Scale bar: $5\times$ standard deviation (SD)).

**Synthetic Data.** We run NCP and vGMFM on 20 sets of synthetic test data each with 500, 1000, and 2000 spikes. As the ground-truth cluster labels are known, we compared the clustering quality using Adjusted Mutual Information (AMI) (Vinh et al., 2010). The AMI of NCP is on average 11% higher than vGMFM (Figure 13), showing better performance of NCP on synthetic data.

**Real Data.** We run NCP, vGMFM and Kilosort on a retina recording with white noise stimulus as described in Appendix D, and extracted the averaged spike template of each cluster (i.e. putative neuron). Example clustering results in Figure 8 (top) shows that NCP produces clean clusters with visually more distinct spike waveforms compared to vGMFM. As real data do not come with ground-truth cluster labels, we compared the spike templates extracted from NCP and Kilosort using retinal

receptive field (RF), which is computed for each cluster as the mean of the stimulus present at each spike. A clearly demarcated RF provides encouraging evidence that the spike template corresponds to a real neuron. Side-by-side comparisons of matched RF pairs are shown in Figure 8 (bottom-left) and Figure 14. Overall, NCP found 103 templates with clear RFs, among which 48 were not found by Kilosort. Kilosort found 72 and 17 of them were not found by NCP (Figure 8 bottom-right), showing that NCP performs at least as well as Kilosort, and finds many additional templates with clear RFs.

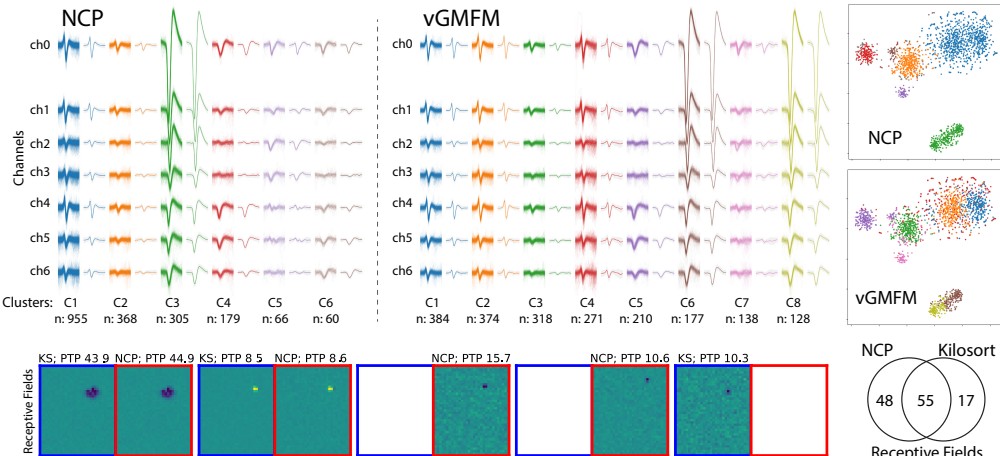

Figure 8: **Spike sorting on real data.** 2000 spikes from real data were clustered by NCP (*top-left*) and vGMFM (*top-mid*). Each column shows the spikes assigned to one cluster (overlaying traces and their average). Each row is one electrode channel. *Top-right:* t-SNE visualization of the spike clusters. *Bottom-left:* Example pairs of matched RFs recovered by NCP (red boxes) and Kilosort (blue boxes). Blank indicates no matched counterpart. *Bottom-right:* Venn diagram of recovered RFs.

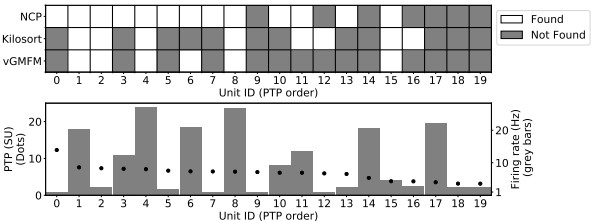

Figure 9: **Spike sorting on hybrid data.** *Top:* NCP, Kilosort, vGMFM recovered 13, 8, and 6 of the 20 injected ground-truth templates. *Bottom:* Peak-to-peak (PTP) size and firing rate of each injected template. (Smaller templates with lower firing rates are more challenging.)

Figure 10: **Clustering ambiguous small spikes.** In both examples, multiple plausible clustering results of small spikes were produced by sampling from the NCP posterior. (scale bar = 5× SD)

**Hybrid Data.** We compared NCP against vGMFM and Kilosort on a hybrid recording with partial ground truth as in Pachitariu et al. (2016). Spikes from 20 ground-truth templates were inserted into a real recording to test the spike sorting performance on realistic recordings with complex background noise and colliding spikes. As shown in Figure 9, NCP recovered 13 of the 20 injected ground-truth templates, outperforming both Kilosort and vGMFM, which recovered 8 and 6, respectively.

**Probabilistic clustering of ambiguous small spikes.** Sorting small spikes has been challenging due to the low SNR and increased uncertainty of cluster assignment. By efficient GPU-parallelized posterior sampling of cluster labels, NCP is able to handle the clustering uncertainty by producing multiple plausible clustering configurations. Figure 10 shows examples where NCP separates spike clusters with amplitude as low as 3-4× the standard deviation of the noise into plausible units that are not mere scaled version of each other but have distinct shapes on different channels.

Overall, our results show that using NCP for spike sorting provides high clustering quality, matches or outperforms a state-of-the-art method, and handles clustering uncertainty by efficient posterior sampling, demonstrating substantial promise for incorporating NCP into production-scale pipelines.

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

# A    NEURAL CLUSTERING PROCESS FOR EXPONENTIAL FAMILIES

The likelihood for an exponential family is given by

$$
\begin{aligned}
p(x|\mu) &= e^{\mu \cdot t(x) - \psi(\mu)} m(x_i) & (12) \\
&= e^{\lambda \cdot h(x)} m(x_i) & (13)
\end{aligned}
$$

where $t(x)$ is a vector of sufficient statistics, and we defined

$$
\begin{aligned}
h(x) &= (1, t(x)) & (14) \\
\lambda &= (-\psi(\mu), \mu) & (15)
\end{aligned}
$$

Let us denote by $K$ and $K' \geq K$ the total number of distinct values in $c_{1:n}$ and $c_{1:N}$, respectively. Consider the joint distribution

$$
p(c_{1:N}, \mathbf{x}, \mu) = p(c_{1:N}) p(\mu) \prod_{k=1}^{K'} e^{\lambda_k \cdot \sum_{i:c_i=k} h(x_i)} \prod_{i=1}^{N} m(x_i) \tag{16}
$$

from which we obtain the marginal distributions

$$
\begin{aligned}
p(c_{1:n}, \mathbf{x}) &= \sum_{c_{n+1} \dots c_N} p(c_{1:N}, \mathbf{x}) & (17) \\
&= \sum_{c_{n+1} \dots c_N} \int d\mu \, p(c_{1:N}) p(\mu) \prod_{k=1}^{K'} e^{\lambda_k \cdot (H_k + \sum_{i>n:c_i=k} h(x_i))} \prod_{i=1}^{N} m(x_i) & (18) \\
&= F(H_1, \dots, H_K, h(x_{n+1}), \dots, h(x_N)) \prod_{i=1}^{N} m(x_i) & (19)
\end{aligned}
$$

where we defined

$$
H_k = \sum_{i \leq n, c_i=k} h(x_i) \qquad k = 1 \dots K \tag{20}
$$

and $H_k = 0$ for $k > K$.

Note now that if $p(c_{1:N})$ is constant, all the dependence of $F$ on $c_{1:n}, x_{1:n}$ is encoded in the $H_k$'s, and $F$ is symmetric under separate permutations of the $H_k$'s and the $h(x_i)$'s for $i > n$. Based on these symmetries we can approximate $F$ as

$$
F \simeq e^{f(G,U)} \tag{21}
$$

modulo adding to $f$ any function symmetric on all $x_i$'s, where

$$
\begin{aligned}
G &= \sum_{k=1}^{K} g(H_k) & (22) \\
U &= \sum_{i=n+1}^{N} u(x_i) & (23)
\end{aligned}
$$

In the conditional probability we are interested in,

$$
p(c_n | c_{1:n-1}, \mathbf{x}) = \frac{p(c_{1:n}, \mathbf{x})}{\sum_{c_n} p(c_{1:n}, \mathbf{x})}, \tag{24}
$$

the product of the $m(x_i)$'s in (19) cancels. Similarly, adding to $f$ a function symmetric on all $x_i$'s leaves invariant our proposed approximation

$$
q_\theta(c_n = k | c_{1:n-1}, \mathbf{x}) = \frac{e^{f(G_k, U)}}{\sum_{k'=1}^{K+1} e^{f(G_{k'}, U)}} \qquad k = 1 \dots K + 1. \tag{25}
$$

# B    DETAILS OF THE NEURAL BLOCK PROCESS

Let us recall the generative model in this case,

$$\begin{aligned}
\alpha, N &\sim p(\alpha), p(N) \\
c_1 \dots c_N &\sim p(c_1, \dots, c_N | \alpha) \\
\phi_{k_1, k_2} &\sim \text{Beta}(\alpha, \beta) \quad k_1 \leq k_2 \\
x_{i,j} &\sim \text{Bernoulli}(\phi_{c_i, c_j}), \quad i \leq j, \quad i, j = 1 \dots N
\end{aligned} \tag{26}$$

where $k_1, k_2 = 1 \dots K$. The prior $p(c_{1:n} | \alpha)$ can be any exchangeable priors for clustering, and the observations $x_{i,j}$ represent the presence or absence of an edge in a graph of $N$ vertices. We set $\phi_{k_1, k_2} = \phi_{k_2, k_1}$ and $x_{i,j} \equiv x_{j,i}$, and assume for notational convenience that $x_{ij} \in \{+1, -1\}$.

## B.1    ENCODING EACH ROW OF THE ADJACENCY MATRIX

In principle posterior inference in this case can proceed similarly to the clustering case, by considering $N$ particles, each given by a row of the adjacency matrix $\mathbf{x}_i = (x_{i,1}, \dots, x_{i,N})$. But we should be careful when encoding these particles. Consider the situation when the values of $c_{1:n}$ have been assigned. Encoding with a generic function $h(\mathbf{x}_i)$ would ignore the permutation symmetries present among the components of $\mathbf{x}_i$, i.e., the columns of the matrix $x_{i,j}$, as a result of the $c_{1:n}$ assignments. These symmetries are the same three symmetries discussed above for clustering models. Moreover, a fixed function $h(\mathbf{x}_i)$ would not be able to accommodate the fact that the length of $\mathbf{x}_i$ changes with the size $N$ of the dataset.

Suppose that there are $K$ clusters among the $c_{1:n}$, each with $s_k$ elements. In order to simplify the notation, let us assume that the $N - n$ unassigned points all belong to an additional $(K + 1)$-th cluster with $s_{K+1} = N - n$, so we assume $c_{n+1:N} = K + 1$, and we have $\sum_{k=1}^{K+1} s_k = N$ and $s_k = \sum_{j=1}^{N} \delta(c_j = k)$.

Now, in each row $\mathbf{x}_i$, the number $s_k$ of elements in the $k$-th cluster can be split as

$$\begin{aligned}
s_k &= s_{i,k}^- + s_{i,k}^+ \\
s_{i,k}^+ &= \sum_{j=1}^{N} \delta(c_j = k) \delta(x_{i,j} = +1) \\
s_{i,k}^- &= \sum_{j=1}^{N} \delta(c_j = k) \delta(x_{i,j} = -1)
\end{aligned}$$

and note that both $s_{i,k}^-$ and $s_{i,k}^+$ are invariant under the symmetry of permuting the indices within cluster $k$.

**Example:** $N = 5$ and $\mathbf{x}_1 = (+1, +1, -1, +1, +1)$. If four assignments were made $c_1 = c_2 = 1$, $c_3 = c_4 = 2$, then $K = 2$ and $c_5 = 3$, and from $\mathbf{x}_1$ we get $s_{1,1}^+ = 2, s_{1,1}^- = 0, s_{1,2}^+ = 1, s_{1,2}^- = 1, s_{1,3}^+ = 1, s_{1,3}^- = 0$. If we permute the columns 3 and 4, both from cluster $k = 2$, we get $\mathbf{x}_1 = (+1, +1, +1, -1, +1)$, but all the $s_{1,j}^\pm$'s stay invariant.

Additional invariants can be obtained combining $s_{j,k}^+$ and $s_{j,k}^-$ across all rows $\mathbf{x}_j$'s with $c_j = c_i$, such as

$$m_{c_i, k}^+ = \frac{1}{s_{c_i}} \sum_{j: c_j = c_i} s_{j,k}^+ \tag{27}$$

$$v_{c_i, k}^+ = \frac{1}{s_{c_i}} \sum_{j: c_j = c_i} (s_{j,k}^+ - m_{c_i, k}^+)^2 \tag{28}$$

and similarly $m_{c_i, k}^-$ and $v_{c_i, k}^-$. Note that these invariants are the same for all rows $\mathbf{x}_j$ with $c_j = k$. The motivation to consider them is that, if the partition corresponding to $c_{1:n}$ is correct, then for $i \leq n$ and $k \leq K$ we have $n_{i,k}^+ \simeq m_{c_i, k}^+$ since they are both estimators of the latent Bernoulli

parameter $\phi_{c_i,k}$. For the same reason, if the partition is correct and those two estimators of $\phi_{c_i,k}$ are exact, then $v_{c_i,k}^+ \simeq 0$. Similarly for $m_{c_i,k}^-$ and $v_{c_i,k}^-$. Then these values provide learning signals to the network that estimates the probability of the assignments $c_{1:n}$ being correct.

Therefore we propose to encode the components of $\mathbf{x}_i$ belonging to cluster k as

$$r_{i,k} = (s_{i,k}^+, m_{c_i,k}^+, v_{c_i,k}^+, s_{i,k}^-, m_{c_i,k}^-, v_{c_i,k}^-) \quad \in \mathbb{R}^6. \tag{29}$$

In order to preserve the symmetry of the first $K$ labels under permutations, we combine them as

$$t_i \quad \equiv \quad \sum_{k=1}^{K} t(r_{i,k}) \quad \in \mathbb{R}^{d_t} \tag{30}$$

where the encoding function is $t : \mathbb{R}^6 \to \mathbb{R}^{d_t}$. The encoding (29) of the unassigned components $x_{i,n+1:N}$ is kept separate and denoted as $q_i = r_{i,K+1}$.

In summary, each row $\mathbf{x}_i$ of the adjacency matrix is represented by the fixed-dimensional pair $(t_i, q_i) \in \mathbb{R}^{d_t+6}$ in a way that respects the symmetries of the assignments $c_{1:n}$: permutations between members of a cluster, permutations of cluster labels and permutations among unassigned columns.

## B.2 CLUSTERING THE ROWS OF THE ADJACENCY MATRIX

We can proceed now as in regular clustering, encoding each cluster of $\mathbf{x}_i$'s within $c_{1:n}$ as

$$H_k = \sum_{i:c_i=k} h(t_i, q_i) \quad \in \mathbb{R}^{d_h} \qquad k = 1 \dots K, \tag{31}$$

and defining the permutation invariant, fixed-dimensional vectors

$$G \quad = \quad \sum_{k=1}^{K} g(H_k), \tag{32}$$

$$U \quad = \quad \sum_{i=n+1}^{N} u(t_i, q_i). \tag{33}$$

In terms of these quantities, the conditional probabilities are defined as usual as

$$q_\theta(c_n = k | c_{1:n-1}, \mathbf{x}) = \frac{e^{f(G_k, U)}}{\sum_{k'=1}^{K+1} e^{f(G_{k'}, U)}} \tag{34}$$

for $k = 1 \dots K + 1$, with $h_n = h(t_n, q_n)$ and with $G_k$ being the value of $G$ for the different configurations. Compared to the regular clustering case, here we need to learn the additional function $t$. We call our approach Neural Block Process (NBP).

## C   MONITORING GLOBAL PERMUTATION INVARIANCE

As mentioned in Section 5, we must verify the symmetry of the posterior likelihood under global permutations of all the data points. We show such a check in Figure 11.

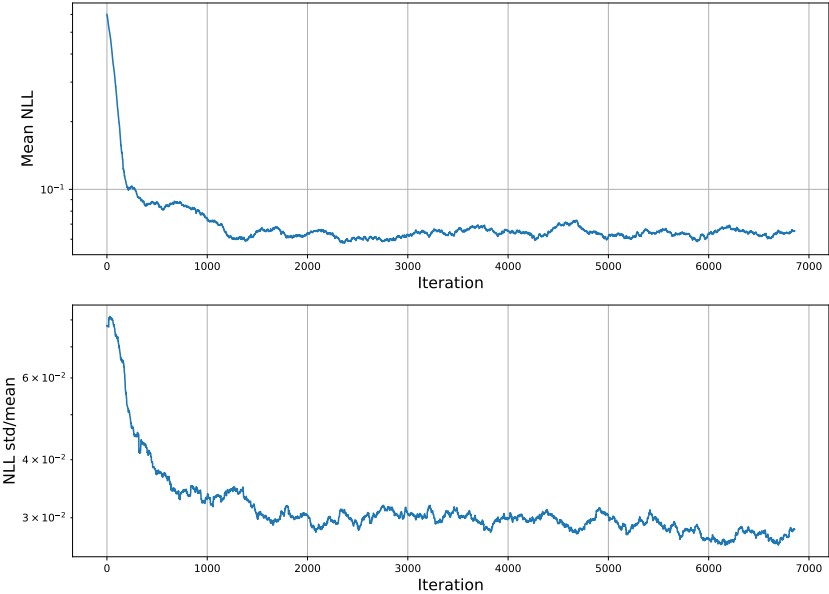

Figure 11: **Global permutation invariance.** Training curves for the NCP model of 2D Gaussians in Section 2. Each minibatch was evaluated for 8 random permutations of the order of the points in the dataset. *Above:* Mean of the NLL over the permutations. *Below:* NLL standard deviation/NLL mean. Note that the ratio is of order $10^{-2}$.

## D   DETAILS OF SPIKE SORTING USING NCP

**Data preprocessing.** Training and test data come from the retinal recordings in Chichilnisky & Kalmar (2002) using a 512-channel 2D hexagonal MEA with 20 kHz sampling rate. After spike detection (Lee et al., 2017), each multi-channel spike waveform was assigned to the channel where the waveform has the maximum peak-to-peak (PTP) amplitude (i.e. the center channel, ch0). This partitioned the recording data by channel such that each center-channel-based partition only contains multi-channel spike waveforms centered at that channel. Each spike waveform is represented as a 7 × 32 array containing the 32 time steps surrounding the peak from the center channel and the same time window from the 6 immediate neighbor channels (Figure 7). These 7 × 32 arrays are the spikes on which clustering was performed.

**Neural architecture for NCP spike sorting.** The overall architecture is the same as the one described in Section 2 and Figure 2. To extract useful features from the spatial-temporal patterns of spike waveforms, we use a 1D ConvNet as the $h$ and $u$ encoder functions. The convolution is applied along the time axis, with each electrode channel treated as a feature dimension. The ConvNet uses a ResNet architecture (He et al., 2016) with 4 residual blocks, each having 32, 64, 128, 256 feature maps (kernel size = 3, stride = [1, 2, 2, 2]). The last block is followed by an averaged pooling layer and a final linear layer. The outputs of the ResNet encoder are the $h_i$ and $u_i$ vectors of NCP, i.e. $h_i = \text{ResNetEncoder}(x_i)$. We used $d_h = d_u = 256$. The other two functions, $g$ and $f$, are identical to those in the 2D Gaussian example.

**Training NCP using synthetic data.** To train NCP for spike clustering, we created synthetic labeled training data using a MFM generative model (Miller & Harrison, 2018) of noisy spike waveforms that mimic the distribution of real spikes:

$$N \sim \text{Uniform}[N_{min}, N_{max}] \quad (35) \qquad c_1 \ldots c_N \sim \text{Cat}(\pi_1, \ldots, \pi_K) \qquad (38)$$

$$K \sim 1 + \text{Poisson}(\lambda) \qquad (36) \qquad \mu_k \sim p(\mu) \quad k = 1 \ldots K \qquad (39)$$

$$\pi_1 \ldots \pi_K \sim \text{Dirichlet}(\alpha_1, \ldots, \alpha_K) \quad (37) \qquad x_i \sim p(x_i | \mu_{c_i}, \Sigma_s \otimes \Sigma_t) \quad i = 1 \ldots N \quad (40)$$

Here, $N$ is the number of spikes between $[200, 500]$. The number of clusters $K$ is sampled from a shifted Poisson distribution with $\lambda = 2$ so that each channel has on average 3 clusters. $\pi_{1:K}$ represents the proportion of each cluster and is sampled from a Dirichlet distribution with $\alpha_{1:K} = 1$. The training spike templates $\mu_k \in \mathbb{R}^{7 \times 32}$ are sampled from a reservoir of 957 ground-truth templates not present in any test data, with the temporal axis slightly jittered by random resampling. Finally, each waveform $x_i$ is obtained by adding to $\mu_{c_i}$ Gaussian noise with covariance given by the Kronecker product of spatial and temporal correlation matrices estimated from the training data. This method creates spatially and temporally correlated noise patterns similar to real data (Figure 12). We trained NCP for 20000 iterations on a GPU with a batch size of 32 to optimize the NLL loss by the Adam optimizer (Kingma & Ba, 2015). A learning rate of 0.0001 was used (reduced by half at 10k and 17k iterations).

**Probabilistic spike clustering using NCP.** At inference time, we fed the 7 x 32 arrays of spike waveforms to NCP, and performed GPU-parallelized posterior sampling of cluster labels (Figure 2 and Figure 7). Using beam search (Graves, 2012; Sutskever et al., 2014) with a beam size of 150, we were able to efficiently sample 150 high-likelihood clustering configurations for 2000 spikes in less

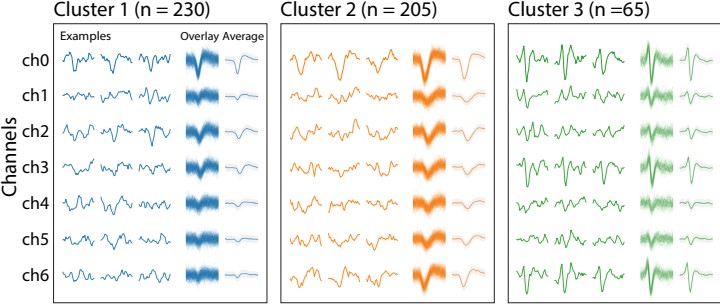

Figure 12: **Synthetic data examples.** Example of 500 synthetic spikes from 3 clusters.

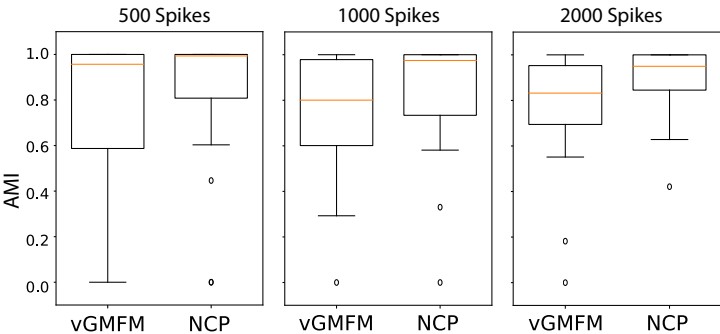

Figure 13: **Clustering synthetic data.** The AMI scores for clustering 20 sets of 500, 1000, and 2000 unseen synthetic spikes.

than 10 seconds on a single GPU. After clustering, we obtained a spike template for each cluster as the average shape of the spike waveforms. The clustering configuration with the highest probability was used for most experiments.

**The spike sorting pipelines for real and hybrid data.** The real data is a 49-channel, 20-minute retina recording with white noise stimulus. To create the hybrid test data, 20 ground-truth spike templates were manually selected from a 49-channel test recording and inserted into another test dataset according to the original spike times.

For NCP and vGMFM, we performed clustering on 2000 randomly sampled spikes from each channel (clusters containing less than 20 spikes were discarded), and assigned all remaining spikes to a cluster based on the L2 distance to the cluster centers. Then, a final set of unique spike templates were computed, and each detected spike was assigned to one of the templates. The clustering step of vGMFM uses the first 5 PCA components of the spike waveforms as input features. For Kilosort, we run the entire pipeline using the Kilosort2 package (Pachitariu, 2019). After extracting spike templates and RFs from each pipeline, we matched pairs of templates from different methods by L-infinity distance and pairs of RFs by cosine distance.

**Electrode drift in real MEA data.** The NCP spike sorting pipeline described above does not take into consideration electrode drift over time, which is present in some real recording data. As a step towards addressing the problem of spike sorting in the presence of electrode drift (Calabrese & Paninski, 2011; Shan et al., 2017), we describe in Sup. Material E a generalization of NCP to handle data in which the per-cluster parameters (e.g. the cluster means) are nonstationary in time.

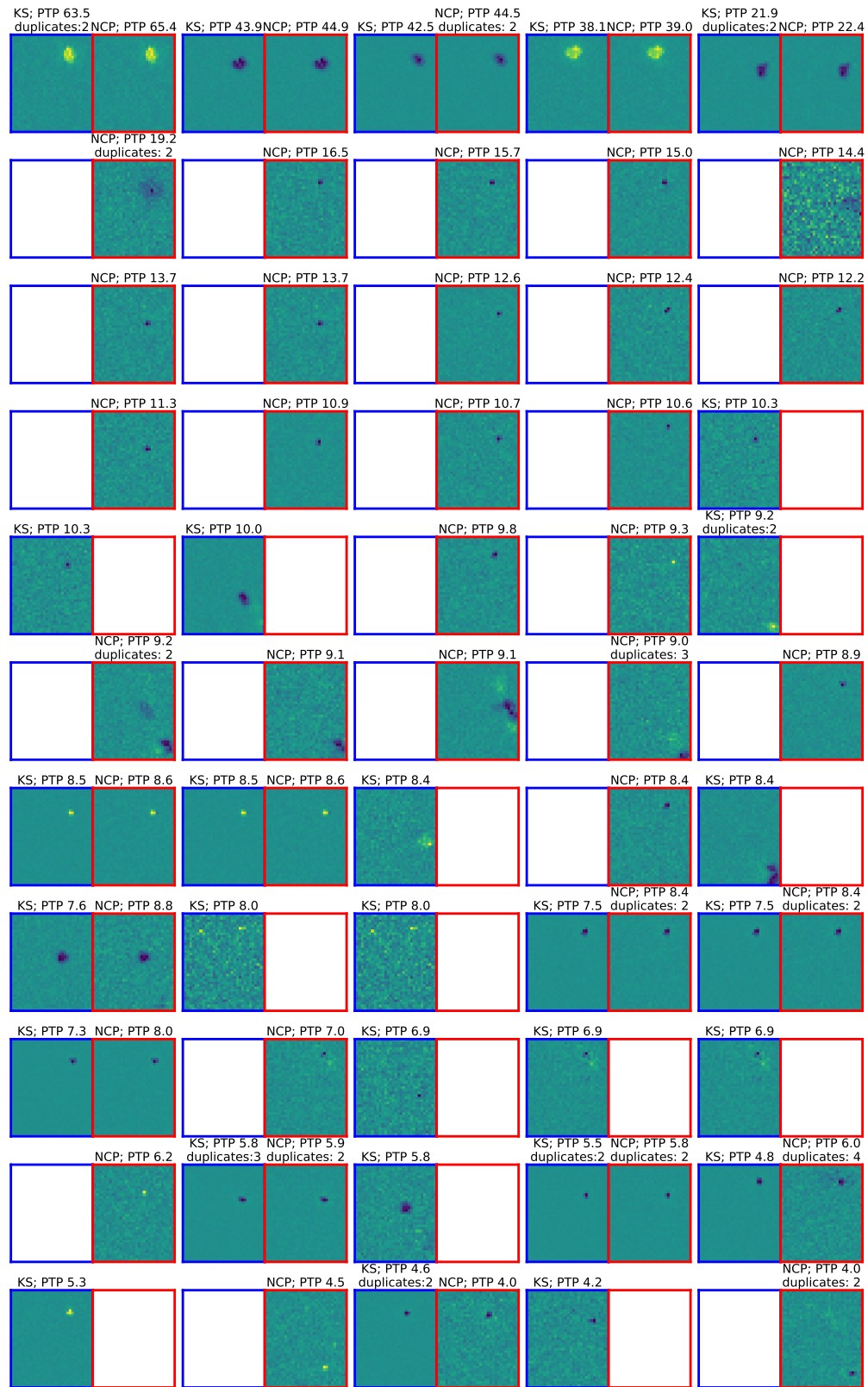

Figure 14: **Spike sorting on real data.** Receptive fields of 55 randomly selected pairs of units recovered from Kilosort and NCP spike sorting. (Red boxes indicate units found by NCP; blue boxes by Kilosort.) Both approaches find the spikes with the biggest peak-to-peak (PTP) size. For smaller-PTP units often one sorting method finds a cell that the other sorter misses. NCP and KS find a comparable number of units with receptive fields here, with NCP finding a few more than KS; see text for details.

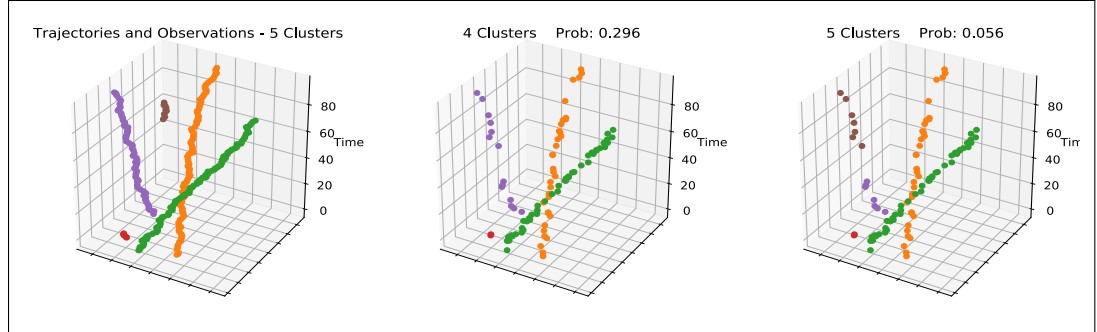

Figure 15: **Neural Particle Tracking.** *Left:* Time trajectories of 5 2D particles. Note that particles can appear or disappear at arbitrary times. *Middle and right:* Two posterior samples. Note that since only one particle is observed at each time, a particle not observed for some time leads to a possible ambiguity on the number of particles. (Best seen in color.)

## E  PARTICLE TRACKING

Inspired by the problem of electrode drift (Calabrese & Paninski, 2011; Pachitariu, 2019; Shan et al., 2017), let us consider now a generative model given by

$$
\begin{align}
c_t &\sim p(c_t|c_1,\ldots,c_{t-1}) & t &= 1,\ldots,T \tag{41}\\
\mu_{k,t} &\sim p(\mu_{k,t}|\mu_{k,t-1}) & k &= 1\ldots K & t &= 1,\ldots,T \tag{42}\\
x_t &\sim p(x_t|\mu_{c_t,t}) & t &= 1,\ldots,T \tag{43}
\end{align}
$$

In this model, a cluster corresponds to the points along the time trajectory of a particle, and (42) represents the time evolution of the cluster parameters. The cluster labels $c_t$ indicate which particle is observed at time $t$, and note that particles can in principle appear or disappear at any time.

To take the time evolution into account, we let particles influence one another with a weight that depends on their time distance. For this, let us introduce a time-decay constant $b > 0$, and generalize the NCP equations to

$$
H_{k,t} = \sum_{t'=1:c_{t'}=k}^{t} e^{-b|t-t'|} h(x_{t'}) \qquad k = 1\ldots K\,, \tag{44}
$$

$$
G_t = \sum_{k=1}^{K} g(H_{k,t})\,, \tag{45}
$$

$$
U_t = \sum_{t'=t+1}^{T} e^{-b|t-t'|} u(x_{t'})\,. \tag{46}
$$

The conditional assignment probability for $c_t$ is now

$$
q_\theta(c_t = k|c_{1:t-1}, \mathbf{x}) = \frac{e^{f(G_{k,t}, U_t)}}{\sum_{k'=1}^{K+1} e^{f(G_{k',t}, U_t)}} \tag{47}
$$

for $k = 1\ldots K+1$. The time-decay constant $b$ is learnt along with all the other parameters. We can also consider replacing $e^{-b|t-t'|}$ with a general distance function $e^{-d(|t-t'|)}$. Figure 15 illustrates this model in a simple 2D example. We call this approach Neural Particle Tracking.

## F  NEURAL ARCHITECTURES IN THE EXAMPLES

To train the networks in the examples, we used stochastic gradient descent with Adam (Kingma & Ba, 2015), with learning rate $10^{-4}$. The number of samples in each mini-batch were: 1 for $p(N)$, 1 for $p(c_{1:N})$, 64 for $p(\mathbf{x}|c_{1:N})$. The architecture of the functions in each case were:

CLUSTERS: 2D GAUSSIANS

- $h$: MLP [2-256-256-256-128] with ReLUs
- $u$: MLP [2-256-256-256-128] with ReLUs
- $g$: MLP [128-256-256-256-256] with ReLUs
- $f$: MLP [384-256-256-256-1] with ReLUs

CLUSTERS: MNIST

- $h$: 2 layers of [convolutional + maxpool + ReLU] + MLP [320-256-128] with ReLUs
- $u$: same as $h$
- $g$: MLP [256-128-128-128-128-256] with ReLUs
- $f$: MLP [384-256-256-256-1] with ReLUs

COMMUNITIES: IRL

- $t$: MLP [6-64-64-64-256] with ReLUs
- $h$: MLP [262-64-64-64-256] with ReLUs
- $u$: MLP [262-64-64-64-256] with ReLUs
- $g$: MLP [256-64-64-64-256] with ReLUs
- $f$: MLP [512-64-64-64-64-1] with ReLUs

