# OpenReview forum: "Neural Clustering Processes"
_ICLR.cc/2020/Conference — Reject_

### Official Review · AnonReviewer3 · 2019-10-23
**Official Blind Review #3**

**Rating:** 6

**Review:**

Summary:
This paper introduces a novel deep learning architecture for efficient amortized Bayesian inference over mixture models. Unlike previous approaches to amortized clustering, the proposed method allows us to treat local discrete labels of data points and infer the unbounded number of mixture components, making it more flexible as in the case of Bayesian nonparametrics. It is shown that the resulting algorithm can be parallelized and applied to both conjugate and non-conjugate models. The authors also suggest an extension to models of random communities and a novel approach to neural spike sorting for high-density multielectrode arrays based on the proposed method.

Strengths:
The paper is generally well written and the relationship to previous works is well described. Empirical results seem quite convincing, for example, the clustering results presented in Fig. 2 and Fig. 3 clearly show not only the inferred number of clusters, but also the posterior probability which indicates that reasonable samples are assigned higher probability.

Weaknesses:
- Overall, the idea looks very original and promising, but I find some technical details are not easy to understand under the current form, especially for non-experts in this domain. I would recommend the authors to elaborate a bit more on the proposed architecture and the variable-input soft-max function in Sect. 2.1.
- On page 8, the authors mention that the NCP is much more efficient compared with MCMC, for example, in the Gaussian 2D example. However, regarding the DPMM clustering model, it is known that MCMC methods are generally slower compared with variational inference, which is computationally faster. I think it would be interesting to add a discussion or comparison with variational inference in terms of computational efficiency.
- If I understand correctly, the NCP is essentially based on a sequential sampling procedure. The authors claim that the proposed method is easily parallelized using a GPU, but there does not seem to be sufficient details on the GPU-parallelization of sequential sampling.

Minor comments:
The size of some figures appears too small, for example Fig. 6 and Fig. 10, which may hinder readability.

**Experience Assessment:**

I have read many papers in this area.

**Review Assessment: Checking Correctness Of Derivations And Theory:**

I assessed the sensibility of the derivations and theory.

**Review Assessment: Checking Correctness Of Experiments:**

I assessed the sensibility of the experiments.

**Review Assessment: Thoroughness In Paper Reading:**

I read the paper at least twice and used my best judgement in assessing the paper.

---

> ### Author Response · Authors · 2019-11-11
> **Reply to Reviewer 3**
>
> Dear Reviewer 3,
>
> Thanks for your comments, which we address below.
>
> 1) Architecture and variable-input soft-max function
>
> We have included a new figure (Figure 1) to help visualize how NCP encodes different cluster configurations.
>
> 2) Efficiency of NCP versus variational inference
>
> In the spike-sorting application we compare the results of NCP versus a variational inference algorithm. In the updated version, we have included in Section 4 an explicit comparison of computational efficiency.
>
> 3) Sequential sampling vs GPU parallelization
>
> The notions of sequential sampling vs GPU parallelization in the paper do not refer to the same task.
>
> A full pass over a dataset of size N can only be performed sequentially by visiting each data point in some order, with a cost O(NK).
>
> But in general we want to perform many O(NK) passes over the same data set, since each pass might give a different result, thus allowing NCP to quantify uncertainty in cluster assignments and number of clusters (this is the primary problem that NCP aims to solve). GPU parallelization performs many such passes in parallel. Each thread still visits sequentially all the data points, but collects a separate set of sampled labels c_n. This only requires a simple modification of Algorithm 1 to include thread indices, and is easy to implement in frameworks such as Pytorch or Tensorflow, as shown in the code we made available.  We have clarified this point in Section 2.1.
>
> 4) We have enlarged the fonts in the indicated figures.

---

### Official Review · AnonReviewer2 · 2019-10-24
**Official Blind Review #2**

**Rating:** 6

**Review:**

The paper presents a neural network based clustering process where the number of clusters is not known a-priori.  Proposed approach requires conjecturing a generative process (where number of clusters/classes is a random variable) and the model learns to uncover the posterior distribution over clusters given the observed samples.
Overall I think it is a valuable contribution, well written paper with good results.
Specific comments:
a) Even though the model allows for variable number of clusters, I feel there may be a strong dependence between the number of clusters the model can hypothesize and the number of clusters in the training data.  It will be useful to give further insights into this.  For instance, if an MNIST model is trained with only digits 0-5 training data, how well would it perform in detecting all 10 clusters at test time?  Understanding model’s biases based on training data is one area I feel is important and the paper could add to.
b) The neural clustering process could potentially be viewed as a transductive inference model for classification of test data.  Typically at test time classification is done for each test sample independently, and the clustering process allows one to bring in other similar test samples to help with classification.  Have the authors considered this and have any comments on potential value / feasibility of this?
c) In the examples presented in Section 2.3, please clarify how training & testing was done.  Specifically what training data was used (all of MNIST training data?), and the test time clustering was done on a subset of MNIST test data?
d) Use of ‘q’ for a neural-network and ‘q_\theta’ for posterior distribution is a little confusing, will be better to have different notation for these.


**Experience Assessment:**

I have published in this field for several years.

**Review Assessment: Checking Correctness Of Derivations And Theory:**

I assessed the sensibility of the derivations and theory.

**Review Assessment: Checking Correctness Of Experiments:**

I carefully checked the experiments.

**Review Assessment: Thoroughness In Paper Reading:**

I read the paper thoroughly.

---

> ### Author Response · Authors · 2019-11-11
> **Reply to Reviewer 2**
>
> Dear Reviewer 2,
>
> Thanks for your comments, which we address below.
>
> a) Distribution shift between training and test data.
>
> We do not expect our method to generalize to datasets from a different distribution than that used for training. Such a generalization is not expected to occur in any generative or discriminative model (unless a model is explicitly designed for transfer learning). Our method assumes the user has already settled on a particular generative model (DPMM, mixture of Gaussians, etc), and provides an efficient way to perform fully Bayesian, amortized inference in such a model. In particular, the marginal distribution over the number of clusters that the model outputs coincides with that of the prior, as illustrated in Geweke’s test in Figure 4.
>
> b) Transductive inference.
>
> In principle one can add additional unlabeled points (either from the training or test sets) as part of the input to q(c|x). Those points would remain as part of the unassigned/marginalized encoding during training, but their presence can potentially improve the likelihood.
>
> c) & d) Train/Test data and notation
>
> In the updated version, we have clarified how training and test data are used in the examples. For the encoding of unassigned points, we have changed the notation from ‘q(x)’ to ‘u(x)’  to avoid confusion with the posterior ‘q(c|x)’.

---

### Official Review · AnonReviewer1 · 2019-10-29
**Official Blind Review #1**

**Rating:** 3

**Review:**


In this paper, the authors consider the neural amortized inference for clustering processes, in which the number of cluster can be automatically adapted based on the observed samples. The proposed algorithm largely follows the standard variational auto-encoder. The major contribution of the paper is the design of the posterior parametrization so that the posterior satisfies the permutation invariant within a cluster, between clusters, and unassigned data, based on the DeepSet method. The model can be incorporated into random communities models. Finally, the authors apply the algorithm for neural spike sorting problem.


The paper is well-organized and easy to follow. However, there are two issues should be addressed:

1, The novelty of the proposed algorithm might not enough. The two major components in this paper, i.e., VAE and DeepSet, are all carefully investigated before. This paper applies the DeepSet parameterization in the VAE framework.

2, The details of the amortized inference training is not clearly explained. It is well-known that gradient through the discrete random variable is quite difficult. How the gradient for the parameter of the proposed model is calculated should be carefully discussed. The REINFORCE gradient in this model, whose support of c can be as large as the number of samples, can be quite huge.

3, In the empirical evaluation, I was curious why the mean-field and MCMC have not been considered in the spike sorting problem.

I am expecting the authors can address my concerns during rebuttal.

=======================================================================

Thanks for the responses to clarify my concerns.

The learning procedure and experiments are clear now.  Indeed, as a purely variational inference paper,  the discrete variable problem is absent, since the model is always *fixed* and not updated.

The major contribution of the paper becomes the design of the posterior parametrization by DeepSets, which I think still not enough.

I will keep my score.

**Experience Assessment:**

I have published in this field for several years.

**Review Assessment: Checking Correctness Of Derivations And Theory:**

I assessed the sensibility of the derivations and theory.

**Review Assessment: Checking Correctness Of Experiments:**

I assessed the sensibility of the experiments.

**Review Assessment: Thoroughness In Paper Reading:**

I read the paper at least twice and used my best judgement in assessing the paper.

---

> ### Author Response · Authors · 2019-11-11
> **Reply to Reviewer 1**
>
> Dear Reviewer 1,
>
> Thanks for your comments, which we address below.  We believe that most of the critical comments here are due to a mis-reading of the paper.  We hope that on a second read these issues are clearer.
>
> “1, The novelty of the proposed algorithm might not enough. The two major components in this paper, i.e., VAE and DeepSet, are all carefully investigated before. This paper applies the DeepSet parameterization in the VAE framework.”
>
> This description is not correct. Our method does not involve a variational autoencoder (VAE), as we clearly stated in the introduction (page 2, paragraphs 2-3). In a VAE one learns both the generative model p(x|z) and the inference model q(z|x) (x observed and z latent). Our work corresponds instead to a situation in which p(x|z) is known or postulated, and we only learn q(z|x). And our novel contribution is a neural network that performs fully Bayesian nonparametric clustering; this has not been done before.  To solve this challenging problem we introduce a novel neural architecture that allows to parametrize q(z|x) when the latents z are cluster labels over arbitrary numbers of clusters for sets of arbitrary size.
>
> “2, The details of the amortized inference training is not clearly explained. It is well-known that gradient through the discrete random variable is quite difficult. How the gradient for the parameter of the proposed model is calculated should be carefully discussed. The REINFORCE gradient in this model, whose support of c can be as large as the number of samples, can be quite huge.”
>
> The well-known challenges of applying gradients through discrete variables are absent in our model. They are relevant in the VAE case, as we point out in page 2, end of paragraph 2. The difficulty arises because the ELBO contains an expectation with respect to q(z|x), so ELBO derivatives must act on that density.
>
> Again, our case is not a VAE, so these problems are not present. Note that our objective function is not an ELBO. Instead, as explained in Section 2.2, our objective is the expectation under p(x) of the KL divergence between the true posterior p(z|x) and its proposed parametrized approximation q(z|x). This leads to an objective function whose form is the expectation of log q(z|x) w.r.t. p(x,z). Since the latter contains no parameters to learn, the derivatives do not act on p(x,z), and there is no problem of gradients over generators of discrete variables. We have added a sentence in Section 2.2 to make this point clear.
>
>  “3, In the empirical evaluation, I was curious why the mean-field and MCMC have not been considered in the spike sorting problem.”
>
> We actually did consider a mean-field approach in the spike-sorting problem. As mentioned in Section 6, paragraph 3, one of the two methods against which we compared NCP is variational inference on a mixture of Gaussians with a Mixture of Finite Mixtures (MFM) prior. We called this method vGMFM. We did not compare here against MCMC because it is not a popular approach to spike-sorting (because it is well-known to be too slow), so we opted for comparing against fast (vGMFM) and state-of-the-art (Kilosort) methods. However, we did compare the runtime of MCMC vs. our method in the 2D Gaussian example, see Section 4.
>
> We hope this clarifies the reviewer’s concerns and provides a fresh perspective on our work. We are happy to clarify any further questions.

---

### Decision · Program_Chairs · 2019-12-19

**Decision:**

Reject

**Comment:**

This paper uses neural amortized inference for clustering processes to automatically tune the number of clusters based on the observed data. The main contribution of the paper is the design of the posterior parametrization based on the DeepSet method.  The reviewers feel that the paper has limited novelty since it mainly follows from existing methodologies. Also, experiments are limited and not all comparisons are made.